# Human Cytomegalovirus Seropositivity and Viral DNA in Breast Tumors Are Associated with Poor Patient Prognosis

**DOI:** 10.3390/cancers14051148

**Published:** 2022-02-23

**Authors:** Zelei Yang, Xiaoyun Tang, Maria Eloisa Hasing, Xiaoli Pang, Sunita Ghosh, Todd P. W. McMullen, David N. Brindley, Denise G. Hemmings

**Affiliations:** 1Department of Biochemistry, University of Alberta, Edmonton, AB T6G 2S2, Canada; zelei@ualberta.ca (Z.Y.); xtang2@ualberta.ca (X.T.); 2Cancer Research Institute of Northern Alberta, University of Alberta, Edmonton, AB T6G 2S2, Canada; todd.mcmullen@albertahealthservices.ca; 3Department of Laboratory Medicine and Pathology, University of Alberta, Edmonton, AB T6G 2B7, Canada; hasing@ualberta.ca (M.E.H.); xiao-li.pang@albertapubliclabs.ca (X.P.); 4Public Health Laboratory, Alberta Precision Laboratory, Edmonton, AB T6G 2J2, Canada; 5Medical Oncology, Mathematical and Statistical Sciences, University of Alberta, Edmonton, AB T6G 1Z2, Canada; sunita.ghosh@albertahealthservices.ca; 6Department of Surgery, University of Alberta, Edmonton, AB T6G 2R7, Canada; 7Medical Microbiology and Immunology, Obstetrics and Gynecology, Women and Children’s Health Research Institute, Li Ka Shing Institute of Virology, Cardiovascular Research Centre, University of Alberta, Edmonton, AB T6G 2S2, Canada

**Keywords:** relapse-free survival, metastasis, tumor recurrence, PCR methods, glycoprotein B, oncomodulatory, latency

## Abstract

**Simple Summary:**

Human cytomegalovirus (HCMV) infects 40–70% of adult populations in developed countries and this is thought to be involved in breast cancer progression; however, reports of detection of the viral genome in breast tumors ranges from 0–100%. We optimized a method that is both sensitive and specific to detect HCMV DNA in tissues from Canadian breast cancer patients. Only ~42% of HCMV-seropositive patients expressed viral DNA in their breast tumors. Viral transcription was not detected in any HCMV-infected breast tumors, indicating a latent infection; however, HCMV seropositivity and the presence of latent infections in breast tumors were independently, and in combination, associated with increased metastasis. HCMV DNA-positive tumors were also associated with lower relapse-free survival. Therefore, HCMV infection status should be accounted for during the monitoring and treatment of breast cancer patients. Prevention or reducing the effects of HCMV infection could decrease morbidity and mortality from metastatic disease.

**Abstract:**

Human cytomegalovirus (HCMV) infects 40–70% of adults in developed countries. Detection of HCMV DNA and/or proteins in breast tumors varies considerably, ranging from 0–100%. In this study, nested PCR to detect HCMV glycoprotein B (gB) DNA in breast tumors was shown to be sensitive and specific in contrast to the detection of DNA for immediate early genes. HCMV gB DNA was detected in 18.4% of 136 breast tumors while 62.8% of 94 breast cancer patients were seropositive for HCMV. mRNA for the HCMV immediate early gene was not detected in any sample, suggesting viral latency in breast tumors. HCMV seropositivity was positively correlated with age, body mass index and menopause. Patients who were HCMV seropositive or had HCMV DNA in their tumors were 5.61 (CI 1.77–15.67, *p* = 0.003) or 5.27 (CI 1.09–28.75, *p* = 0.039) times more likely to develop Stage IV metastatic tumors, respectively. Patients with HCMV DNA in tumors experienced reduced relapse-free survival (*p* = 0.042). Being both seropositive with HCMV DNA-positive tumors was associated with vascular involvement and metastasis. We conclude that determining the seropositivity for HCMV and detection of HCMV gB DNA in the breast tumors could identify breast cancer patients more likely to develop metastatic cancer and warrant special treatment.

## 1. Introduction

Breast cancer is the most common malignancy among women worldwide and it is their leading cause of death from cancer [1]. Examples of the risk factors for breast cancer development include aging, family history, genetic mutations, reproductive factors such as pregnancy and menopause, endogenous and exogenous effects of estrogen, unhealthy lifestyle such as drinking and smoking, high body-mass index and dense breast tissue [2,3,4,5,6]. Another potential risk factor for the development and progression of breast cancer is viral infection.

Infectious agents contribute to >15% of all cancers and ~64% of these agents are viruses [7]. Viruses with causative effects in cancer include the Epstein–Barr virus leading to Burkitt’s lymphomas [8,9] and human papillomavirus leading to cervical cancer [10,11]. Some recent studies also show that human cytomegalovirus (HCMV) has oncogenic potential where it directly leads to oncogenesis [12,13,14]. Indeed, the virus is found in cancerous tissues, including colorectal [15], prostate [16], ovarian [17] and glioblastoma cancers [18]. Increasing evidence also shows that HCMV infection plays a role in breast cancer [19,20,21]. This may be oncogenic [12,13,14] or oncomodulatory where the infection contributes to cancer progression through the alteration of intracellular signaling pathways [22,23].

CMV is a β-herpesvirus with double-stranded DNA and infection is highly host specific [24]. HCMV produces lifelong infections in 40–70% of adult populations in developed countries [25,26] and this increases to 85–90% when people reach 75–80 years of age [27]. A higher HCMV seroprevalence is observed in South America, Africa and Asia, where >80% of the non-elderly adult population is infected [28]. Transmission of HCMV is commonly through direct contact with body fluids such as blood, saliva, breast milk and organ transplantation, into which infectious viral particles can be shed during an active infection [29,30,31]. HCMV infection normally causes no noticeable symptoms in immunocompetent individuals; however, it can be associated with fever, malaise, abnormal liver function and infectious mononucleosis [32,33,34]. After active replication of the virus, the infection will enter a state of latency where there is restricted expression of viral genes and limited viral production [35]. Viral reactivation and production of an infectious virus can occur during inflammation, stress, in the aging population, and in immunocompromised hosts [36,37,38,39]. The infected individual is seropositive for HCMV immunoglobulin G (IgG) and the latency stage ensures that this virus is never fully eliminated [40].

The presence of HCMV proteins, DNA and/or mRNA in patient breast tumors have been examined by several investigators, but the results are controversial. HCMV immediate early (IE), early and/or late proteins are detected in 90–100% of human breast tumors in some studies [19,41,42,43,44], but at lower incidences in others [45,46,47]. Studies that detect HCMV proteins in a high proportion of breast tumors also show >90% detection of HCMV DNA in these same tumors [19,41,42,43], with two studies also detecting mRNA for HCMV IE [42,43]. By contrast, some studies show a 40–80% positivity of HCMV DNA detection [19,48,49,50], while others report very few or no HCMV DNA-positive breast tumors [45,47,51,52,53,54,55,56,57]. These differences could relate to the variability in serostatus for HCMV in the populations studied, which ranges from 60–100% [45,52,57,58]; however, in a population that is >90% HCMV seropositive, the detection of HCMV DNA in breast cancerous tissues is found to be as low as <20% [45,57]. Alternatively, these discrepancies in the detection of HCMV in breast tumors could also be due to variability in the cellular composition of the breast tumors. We recently showed that the incidence of HCMV infection of breast cancer cells is considerably less than that of fibroblasts with higher infection dependent on the expression of platelet-derived growth factor receptor alpha (PDGFRα) [59]. Fibroblasts are a component of the microenvironment and they are likely to be infected more than the breast cancer cells.

Several methods were also used for the detection of HCMV DNA in breast tumors, including in situ hybridization and real-time, standard or nested PCR. The use of these different detection methods, the detection of different viral genes and the source of the starting material such as fresh-frozen or paraffin-embedded tissues could offer another explanation for these disparities [60]. Therefore, it is critical to identify the optimum assay for a chosen viral gene that is both sensitive and specific for detecting HCMV DNA.

The association between HCMV infection and outcomes from breast cancer is also important. HCMV DNA and/or proteins in breast tumors have been associated with higher tumor grade [46,49], invasive breast cancer [49], and negative expression of the estrogen receptor-1 (ER), progesterone receptor (PR) and human epidermal growth factor receptor 2 (HER2) [42,46]. Positive detection of HCMV DNA in breast tumors is also associated with poor overall and low relapse-free survival [61]. HCMV DNA and/or proteins are present in nearly all the metastatic lymph nodes [41,43] and brain tissues [62] of breast cancer patients. Our previous work with mouse models of breast cancer showed that the latent infection of mice with mouse cytomegalovirus (mCMV) increases lung metastases, where mCMV DNA, but not mRNA, was detected in the lung metastatic nodules [63]. These combined results indicate that latent CMV infection is associated with worse patient outcomes; however, the relationship between HCMV seropositivity and the presence of HCMV DNA in breast tumors is unclear because the serostatus of patients in previous studies was either unknown or almost all patients were infected. Understanding this relationship and how each of these parameters correlate with patient outcomes is especially important because of the prevalence of the population infected with HCMV, which reaches 40–70% even in developed countries [28]. Testing for seropositivity only requires a blood test and could be useful for directing increased monitoring.

In the present study, we assessed two HCMV DNA detection methods, LightCycler (LC)-PCR and nested PCR. Nested PCR targeting the HCMV *glycoprotein B (gB)* gene provided the required sensitivity and specificity for detecting HCMV DNA in patient breast tumors. Using this method, no significant difference in the detection of HCMV DNA was found in 136 patient breast tumors obtained from a local Canadian population, compared to 10 breast tissue samples from women not diagnosed with breast cancer. The number of breast tumors, which were positive for HCMV DNA, was much lower than the HCMV IgG seropositivity rate in these patients. Notwithstanding, HCMV seropositivity and the presence of gB DNA in the breast tumors were each positively associated with metastasis. Patients with HCMV gB DNA detected in the breast tumors were also at higher risk of having a lower relapse-free survival time. In addition, mRNA of HCMV IE1 was not detected in the HCMV gB DNA-positive breast tumors, indicating a presumably latent infection. Our results, therefore, demonstrate that breast cancer patients who are HCMV seropositive need more intensive monitoring for disease progression, particularly those patients where the excised tumors are positive for HCMV DNA.

## 2. Materials and Methods

### 2.1. Reagents

All common chemicals and reagents were purchased from Sigma-Aldrich (Oakville, ON, Canada) or Thermo Fisher Scientific (Waltham, MA, USA) unless otherwise stated. All primers were from Integrated DNA Technologies Inc. (Coralville, IA, USA).

### 2.2. Human Specimens and Patient Information

Breast tumors from 136 breast cancer patients were obtained after surgical removal and they were immediately frozen in liquid nitrogen, followed by storage at −80 °C by the Alberta Cancer Research Biobank from where they were obtained. Normal breast tissues from ten women who had no history of cancer were obtained from breast reduction surgeries and processed similarly as the breast tumors. Patient samples were obtained with the approval of the University of Alberta Health Research Ethics Board (Pro00018758) with written informed consent.

Matching patient information was available for each breast tumor, including age, menopause status, body mass index, tumor grade, number and size of lymph node metastases, tumor stage, expression of estrogen receptor (ER)/progesterone receptor (PR)/human epidermal growth factor receptor 2 (HER2) in the tumor, tumor recurrence, vascular invasion and patient survival. All information was updated in November 2020. Cases with unknown or missing information in each category were excluded from the statistical analysis for that category. Matching serum samples for 84 of these breast cancer patients were available for serological tests.

### 2.3. Positive and Negative Controls for HCMV Infection

A clinical isolate of HCMV, Kp7 (from Dr. Jutta K. Preiksaitis, Department of Medicine, University of Alberta) [64], was used to generate the HCMV-positive control. Freshly obtained human breast adipose tissue from a breast reduction surgery that initially tested negative for HCMV DNA was actively infected with 1.25 × 10^4^ virus/mL of HCMV in culture for 24 h and then extracted for DNA and RNA. The extracted DNA was serially diluted 10-fold (neat to 10^−5^). The HCMV-negative control was prepared from human MDA-MB-231 breast cancer cells (American Type Culture Collection, Manassas, VA, USA), where the extracted DNA and mRNA were both negative for HCMV.

### 2.4. Extraction of DNA and RNA

DNA and RNA were extracted using the All-In-One DNA/RNA/Protein Miniprep Kit (Bio Basic, Markham, ON, Canada) following the manufacturer’s protocol. Briefly, tissues (~20 mg) were homogenized in a 350 μL Buffer Lysis-DRP contained in 2 mL microcentrifuge tubes with 5-mm stainless steel beads, using the Qiagen TissueLyser II system (24-sample plates, 25 Hz, 5 min) (Qiagen, Toronto, ON, Canada). Tissue lysates were centrifuged at 12,000× *g* for 3 min at 4 °C and the supernatants were transferred into new RNase-Free tubes. Lysates were transferred into the EZ-10 DNA Columns and centrifuged at 9000× *g* for 1 min at room temperature, with the columns kept for DNA purification at room temperature and the flow-throughs transferred to new RNase-Free tubes for RNA purification at 4 °C.

For DNA purification, the EZ-10 DNA columns were washed sequentially by 350 μL Buffer Lysis-DRP, 500 μL CW1 Solution and 500 μL CW2 Solution. Each solution was added to the columns and incubated for 1 min, then centrifuged at 9000× *g* for 1 min before the next solution was added. The columns were further centrifuged at 9000× *g* for 2 min to remove residual liquid. The columns were opened for 3 min to evaporate the ethanol, then incubated with 50 μL CE Buffer for 2 min and centrifuged at 9000× *g* for 2 min to elute the DNA. Samples were stored at −20 °C.

For RNA purification, 250 μL ethanol was added to the previously collected flow-throughs, with the mixture transferred into the RZ-10 RNA columns and centrifuged at 9000× *g* for 1 min. The second flow-throughs were kept for protein purification. The RZ-10 RNA columns were washed sequentially by 500 μL GT Solution and 500 μL NT Solution. Each solution was added to the columns and incubated for 1 min, then centrifuged at 9000× *g* for 1 min before the next solution was added. The columns were further centrifuged at 9000× *g* for 2 min to remove the residual solution. The columns were incubated with 50 μL RNase-Free Water for 2 min and centrifuged at 9000× *g* for 2 min to elute the RNA. Samples were stored at −80 °C.

### 2.5. LightCycler (LC) PCR

The LC-PCR reaction mixture contained 5 μL of DNA, 4 mM MgCl_2_, 0.5 μM of each primer, 0.2 μM of each probe, and 2 μL of the reagent from a LC-FastStart DNA Master hybridization probe kit (Roche Diagnostics, Laval, QC, Canada) for a total reaction volume of 20 μL. The primers used for the detection of HCMV gB were as follows: forward: 5′-TACCCCTATCGCGTGTGTTC-3′ and reverse: 5′-ATAGGAGGCGCCACGTATTCT-3′. The hybridization donor probe with a fluorescein 3′-end label and the acceptor probe with a LC-Red 640 5′-end label were used in the LC-PCR reaction (TIB Molbiol LLC, New Jersey, NJ, USA) [65]. The reaction was performed in the LightCycler 480 Instrument II (Roche Diagnostics, Laval, PQ, Canada), with the following thermal cycling conditions: (1) 10 min at 95 °C and (2) 45 cycles of 15 s of denaturing at 95 °C, 10 s of annealing at 55 °C, and 10 s extension at 72 °C. Measurements were collected during the annealing period with a channel setting F2/F1 for real-time detection of the amplification. The specificity of the fluorescence signal was checked by a melting curve analysis, with *T*_m_ = 67.5 °C for the probes.

A positive standard curve was generated with a series of log dilutions containing 10^6^ to 10^1^ genome copies of HCMV, using purified viral DNA that was quantified by spectrophotometry with absorbance at 260 nm [65]. The LC-PCR program compared the results from the tested samples against the standard curve to determine the number of HCMV genome copies in a sample reaction.

### 2.6. Nested PCR Amplification and Visualization

The presence of HCMV DNA in human tissues was determined using primers specific to either HCMV *IE1* (fourth exon) [58,66] or *gB (UL55)* gene [66]. Each PCR reaction had a 50 μL reaction volume, containing 2X PCR Taq Master Mix (Applied Biological Materials, Richmond, BC, Canada). The extracted DNA (150 ng) was amplified with the external primer set for the first round of amplification, then 2 μL of the amplified product was used with the internal primer set for the second round of amplification. The PCR reaction was performed with an initial denaturation for 4 min at 94 °C, followed by specific thermal cycling conditions that were individually listed for each primer set in Appendix A in the order of denaturation, annealing and elongation, and then a final elongation for 7 min at 72 °C. Optimization of the PCR conditions was performed for the HCMV IE1 primers by testing a range of annealing temperatures. For the external primers, annealing temperatures of 55, 60, 62, 65 and 67 °C were tested, with all second rounds of reaction performed at 50 °C. The internal primers were tested at annealing temperatures of 50, 53 and 55 °C, after the first round of reaction was performed at 62 °C. The products were visualized after separation on an agarose gel, staining with ethidium bromide and exposure to ultraviolet light. For products > 100 base pair (bp), a 1.5% gel was used, and electrophoresis was performed at 100 V for 30 min. For products < 100 bp, a 3% gel was used and developed for 1 h at 100 V. The band size was determined relative to the 100 bp DNA ladder. All steps were performed with care to avoid cross contamination between samples and new aliquots of reagents were used for each round of PCR reaction.

### 2.7. Sequencing of PCR Product

PCR products with the expected size were cut out from the agarose gel for re-extraction of DNA using the MinElute Gel Extraction Kit (Qiagen, Toronto, ON, Canada) following the manufacture’s protocol. PCR products < 100 bp were cloned into vectors for effective DNA sequencing. Cloning was performed using the pUCM-T Cloning Vector Kit (Bio Basic Inc., Markham, ON, Canada) and in competent E. coli cells. The resulting colonies were screened for transformants, with the white colonies representing recombinant clones. Each colony was picked up with a toothpick and placed in 5 mL of LB medium containing ampicillin for overnight incubation at 37 °C. Plasmid DNA was extracted from the liquid culture using the Column-Pure Plasmid Miniprep Kit (Applied Biological Materials, Richmond, BC, Canada). Samples were analyzed using the Sanger DNA Sequencing service at the Molecular Biology Facility (University of Alberta, Edmonton, AB, Canada). The sequence of the target clones was determined by M13 universal forward primer: 5′-GTAAAACGACGGCCAGT-3′. The resulting sequences were aligned to the target gene using the CLC Sequence Viewer software (CLC bio, Aarhus, Denmark) to verify the product specificity. The sequences were further identified using the Basic Local Alignment Search Tool (BLAST) database.

### 2.8. qPCR for mRNA Expression

mRNA was reverse transcribed to complementary DNA (cDNA) using the 5X All-In-One Reverse Transcription MasterMix (Applied Biological Materials, Richmond, BC, Canada). The cDNA was analyzed for the relative amount of target genes by qPCR using EvaGreen qPCR master mix (Applied Biological Materials, Richmond, BC, Canada). The relative abundance of HCMV IE1 [67] and PDGFRα expression at the mRNA level was determined by normalizing against a housekeeping gene, *glyceraldehyde 3-phosphate dehydrogenase (GAPDH).* HCMV IE: sense 5′-TGAGGATAAGCGGGAGATGT-3′ and antisense 5′-ACTGAGGCAAGTTCTGCAGT-3′. PDGFRα: sense 5′- TAGTGCTTGGTCGGGTCTTG -3′ and antisense 5′- TTCATGACAGGTTGGGACCG -3′. GAPDH: sense 5′-TCCTGCACCACCAACTGCTT-3′ and antisense 5′-TCTTACTCCTTGGAGGCCAT-3′.

### 2.9. HCMV IgG Detection

The presence of HCMV IgG antibodies in serum samples was determined qualitatively by using the CMV IgG ELISA kit (Genway Biotech Inc., San Diego, CA, USA) according to the manufacturer’s instructions. Samples were added to microtiter strip wells precoated with CMV antigens, allowing the binding of the CMV IgG antibodies to the well. Wells were washed and horseradish peroxidase (HRP) labelled anti-human IgG conjugate was added. Tetramethylbenzidine, the substrate for HRP was then added, giving a blue product when CMV IgG antibodies were present in the sample. Sulfuric acid was added to stop the reaction, resulting in a yellow endpoint color, with the absorbance determined at 450 nm using Easy Reader EAR 340 AT (SLT-Lab Instruments, Salzburg, Austria). Appropriate controls were included as suggested by the manufacturer to determine the positivity of tested samples.

### 2.10. Statistical Analysis

Descriptive statistics were used to present the study variables. The mean and S.D. or median and interquartile range (IQR) were reported for continuous variables, based on a normal or skewed distribution of the results, respectively. Frequency and proportions were reported for categorical variables. Chi-square tests were used to correlate two categorical variables. Fisher’s exact tests were reported when the cell frequency was less than 5. Binary logistic regression was used to identify the factors associated with the gB DNA (negative versus positive) outcome variable. A univariate binary logistic regression model was used to find the variables associated with the outcome variable. Factors significant at *p* < 0.10 were entered into the multivariate model. The final multivariate model was chosen based on the statistical significance and clinical relevance. A *p*-value < 0.05 was used for statistical significance. SPSS (IBM Corp. Released 2017. IBM SPSS Statistics for Windows, Version 25.0. Armonk, NY, USA: IBM Corp.) version 25 was used for all statistical analysis.

## 3. Results

### 3.1. Comparison of HCMV gB DNA Detection Using LC-PCR versus Nested PCR

We compared the limit of detection for HCMV gB DNA as measured by LC-PCR and nested PCR in parallel. The HCMV DNA-positive control including the original DNA extract and dilutions at 1:10, 1:10^2^, 1:10^3^, 1:10^4^ and 1:10^5^ were tested. The HCMV DNA-negative control was included in each test. The LC-PCR analysis resulted in an estimate of 200 copies of the HCMV viral genome in 10 μL of the original stock of HCMV DNA-positive control based on calculations using a previously established standard curve that was generated with purified HCMV DNA of known genome copies (Table 1) [62]; however, 104 copies of the viral genome were estimated to be present in 10 µL of the 1:10 diluted HCMV DNA-positive control instead of an expected 20 copies (Table 1). LC-PCR was able to detect HCMV gB DNA up to 1:10^2^ dilution of the HCMV positive control, with an estimation of two copies of the viral genome in the reaction, while 1:10^3^, 1:10^4^ and 1:10^5^ dilutions were all below the level of detection (Table 1). By contrast, the nested PCR detected a band at 96 bp on the gel, representing the HCMV *gB* gene, even when the HCMV-positive control was diluted 1:10^5^ (Figure 1). This demonstrates that the nested PCR had an ~1000-fold greater sensitivity in comparison to the LC-PCR; however, different dilutions of the HCMV-positive control resulted in no obvious difference in the intensity of bands, representing saturation of the reactions, where observation of a band would only indicate the presence of the viral gene, but not its level (Figure 1). Both methods showed no detection of HCMV gB DNA in the HCMV-negative control sample (Table 1, Figure 1 and Figure 2). Furthermore, no non-specific bands were observed for the nested PCR product of HCMV gB in the negative HCMV control, indicating that the reaction was specific (Figure 1).

The presence of the HCMV viral genome in human breast tumors and normal breast tissues was examined by nested PCR that targeted either the *gB* or *I**E1* genes, using a subset of the total samples to confirm the methods. The HCMV-positive (1:10^3^ dilution) and negative controls were included in every round of PCR amplification. A reaction mixture with no template was also included to monitor contamination during the experimental procedure. No bands were observed on the gel for any of these negative controls.

The PCR products generated by targeting the HCMV *gB* gene in human breast tumor samples showed no band or a single band of similar intensities at 96 bp similar to the positive control (Figure 2A). The identity of this band as CMV gB DNA was verified by Sanger sequencing of ten clones created from bands extracted from the gels. All ten sequences showed alignment to the UL55 genomic sequence that encodes HCMV *gB*. Four of these clones are illustrated as examples in the sequence alignment map (Figure 2B). This demonstrates that the 96 bp PCR products were not false positives.

On the other hand, the detection of HCMV IE1 DNA in human breast tumor samples showed ambiguous results. As expected, the HCMV-positive control showed a single band at 293 bp on the gel (Figure 3A). Multiple bands were observed in the HCMV negative control, although the products were <200 bp (Figure 3A). The human breast tumor samples showed highly variable results with multiple bands. To increase the primer specificity a gradient of annealing temperatures was tested in the PCR reaction, up to a 5 °C increase in either the external or internal primer reactions, but this still resulted in multiple bands for the human breast tumor samples tested (Appendix A). Some samples appeared to have a product of 293 bp, but this was not the most intense band (Figure 3A). Seven patient samples were chosen randomly, with the observed ~293 bp bands re-extracted for the DNA products. These samples were sequenced but none aligned with the IE1 genomic sequence (Figure 3B). The resulting sequences were identified using the BLAST database and all were found to be from the human genomic sequence. Therefore, the detection of HCMV IE1 using nested PCR produced non-specific products and false-positive results.

### 3.2. Detection of HCMV gB DNA in Tissues and HCMV IgG in Serum

Nested PCR for the detection of HCMV gB DNA was specific and it was, therefore, used to analyze all tissues. Tissue samples from breast reduction surgeries from ten women without breast cancer and 136 breast tumors from patients were analyzed resulting in 10% (1/10) and 18.4% (25/136) positivity for HCMV gB DNA, respectively. These results were not significantly different (*p* = 0.691, Table 2). Of the 84 matched serum samples that were available from the breast cancer patients, 41.7% (35/84) were negative and 58.3% (49/84) were positive for HCMV IgG antibody (Table 2). All patients who were seronegative for HCMV IgG were also negative for HCMV gB DNA in the breast tumors, thus confirming the lack of infection. Conversely, all women who had HCMV gB DNA positive tumors were also positive for serum HCMV IgG. There were ten patients with HCMV gB DNA detected in the breast tumors but for whom matching serum samples were not available. Therefore, for further analyses, we assumed that these ten patients with breast tumors would have also been seropositive for HCMV. As a result, we calculated that 62.8% (59/94) of our patient population were HCMV seropositive and 37.2% (35/94) were HCMV seronegative (Table 2). Including the ten patients with assumed HCMV seropositivity, 42.4% (25/59) of the HCMV IgG positive patients were also positive for HCMV gB DNA in breast tumors, while no HCMV gB DNA was detected in 57.6% (34/59) of HCMV seropositive women (Table 2).

### 3.3. Association of HCMV gB DNA-Positive Breast Tumors or HCMV IgG Positivity with Patient Characteristics

We next examined whether the presence of HCMV gB DNA in breast tumors was associated with any patient characteristics. The average age of all patients was 56, ranging from 23 to 80 years (Table 3). Most patient characteristics did not differ between groups; however, out of all patients that had tumors positive for HCMV gB DNA, 64.0% (16/25) had a tumor recurrence event, which was significantly higher compared to the 43.2% (48/111) of patients in the HCMV gB DNA-negative group in univariate analysis (*p* = 0.06, Table 3). In addition, 32.0% (8/25) of patients with HCMV gB DNA-positive tumors had Stage IV or metastatic breast cancer, which was significantly higher than the 10.8% (12/111) for the HCMV gB DNA-negative group (*p* = 0.007, Table 3). These results showed that HCMV gB DNA positivity in breast tumors was positively associated with tumor recurrence events and metastasis.

We also examined whether being HCMV seropositive, regardless of gB DNA status in breast tumors, was associated with any patient characteristics by univariate analysis. Patients who were positive for HCMV IgG were significantly older at the time of breast cancer diagnosis (57 ± 12 years) compared to those that were HCMV IgG negative (51 ± 13 years) (*p* = 0.017, Table 4). A higher percentage of HCMV IgG negative patients (50.0%, 17/34) were pre-menopausal compared to those that were IgG positive (21.1%, 12/57). Correspondingly, a higher percentage of HCMV IgG positive patients were post-menopausal (68.4%, 39/57) and 10.5% (6/57) were pre-menopausal compared to those that were IgG negative (47.1%, 16/34 or 2.9%, 1/34, respectively). Overall, HCMV IgG seropositivity was significantly associated with post- and peri-menopause (*p* = 0.012, Table 4). In addition, patients who were positive for HCMV IgG had a significantly higher body mass index (29.9 ± 7.36 kg/m^2^) compared to those that were HCMV IgG negative (27.5 ± 6.02 kg/m^2^) (*p* = 0.093, Table 4); however, age, menopause status and body mass index were not significantly associated with the presence of HCMV gB DNA in the breast tumors (Table 3).

Stage IV or metastatic breast cancer was significantly higher in patients who were HCMV IgG seropositive with HCMV gB DNA detected in breast tumors (32.0% (8/25)) compared to patients that were seropositive but HCMV gB DNA-negative (11.8% (4/34)) when assessed by univariate analysis (*p* < 0.0995, Table 5). Similarly, IgG seropositive patients with detection of gB DNA in the breast tumor had significantly increased vascular involvement (70.8% (17/24)) compared to IgG-positive patients with gB DNA-negative tumors (45.5% (15/33)) (*p* = 0.057, Table 5); however, vascular involvement was not associated with either HCMV seropositivity or gB DNA-positive tumors on their own (Table 3 and Table 4).

Other patient outcomes and characteristics including overall survival status, overall time to recurrence event, tumor grade, number and size of metastatic lymph nodes, expression of ER/PR/HER2, tumor subtypes or tumor PDGFRα mRNA expression were not significantly different when assessing patients based on the presence or absence of HCMV gB DNA in breast tumors or based on HCMV serostatus (Table 3, Table 4 and Table 5).

Based on these results from the univariate analyses, we next performed multivariate analysis. The odds of being HCMV gB DNA positive in breast tumors or being HCMV IgG positive regardless of gB DNA status in patients with Stage IV breast cancer was 5.27 or 5.61 times higher, respectively, than patients with Stage I-III (95% CI 1.77–15.67, *p* = 0.003, 95% CI 1.09–28.75, *p* = 0.039; Table 6). HCMV gB DNA positivity was not significantly associated with menopause status even after adjustment (Table 6); however, when compared to pre-menopausal patients, the odds of being HCMV IgG positive regardless of gB DNA status for post-menopause patients was 3.83 times higher (95% CI 1.43–10.29, *p* = 0.008) and for peri-menopause patients was 11.42 times higher (95% CI 1.18–110.31, *p* = 0.035, Table 6). Within the HCMV seropositive group, the odds ratio of having gB DNA-positive breast tumors and being peri- or post-menopausal were not significantly different compared to seropositive patients with gB DNA-negative tumors (Table 6; however, there was a trend towards an increased odds ratio of developing Stage IV breast cancer in seropositive patients with gB DNA-positive tumors compared to those who were gB DNA negative (3.48, 95% CI 0.88–13.78, *p* = 0.076).

### 3.4. HCMV gB DNA Positivity in Breast Tumors was Associated with Reduced Relapse-Free Survival

We examined the association between HCMV gB DNA-positive tumors and survival time. Median overall survival time in patients who were HCMV gB DNA positive (7.06 ± 2.17 years, 95% CI 2.81–11.3) compared to those that were negative (8.72 ± 1.36 years, 95% CI 6.05–11.4) was not significantly different (*p* = 0.614, Figure 4A). Patients that had HCMV gB DNA-negative or -positive tumors had 64.4% versus 59.8% overall survival rates at the five-year interval and 45.6% versus 45.0% at the ten-year interval, respectively (Figure 4A); however, the median relapse-free survival time for patients with HCMV gB DNA-positive tumors (3.45 ± 1.61 years, 95% CI 0.29–6.61) was significantly lower than for those with HCMV gB DNA-negative tumors (8.51 years, 95% CI not reached) (*p* = 0.039, Figure 4B). Patients with HCMV gB DNA-negative or -positive tumors had 58.3% versus 33.6% relapse-free survival rates at the five-year interval and 49.1% versus 28.8% at the ten-year interval, respectively (Figure 4B). The risk of reduced relapse-free survival was 1.8 times higher in patients who were positive for HCMV gB DNA in their tumors compared to those whose tumors were negative (95% CI 1.02−3.17, *p* = 0.042, Figure 4B).

Neither overall nor relapse-free survival were different in patients based solely on HCMV serostatus (Appendix A). The five-year and ten-year overall survival rates for HCMV IgG-negative patients were 70.9% and 53.6%, respectively, and for IgG-positive patients they were 64.1% and 51.6%, respectively. The five-year and ten-year relapse-free survival rates for the HCMV IgG-negative patients were 63.2% and 58.3%, respectively, and for the HCMV IgG-positive patients were 46.3% and 39.1%, respectively. The risk of reduced relapse-free survival was 1.55 times higher in the HCMV IgG-positive patients, but this did not reach statistical significance (95% CI 0.81−2.96, *p* = 0.185, Appendix A).

### 3.5. mRNA Expression of HCMV IE1 Was Not Detected in any Human Breast Tumors or Normal Breast Tissue

The expression of HCMV IE1 mRNA was examined in the same set of human breast tumors (*n* = 136) and normal breast tissues (*n* = 10). The HCMV-positive and -negative controls were included in each round of qPCR analysis. The HCMV-positive control resulted in an amplification cycle number of ~22 and a single peak in the melting curve at 86.5 °C (Appendix A). All of the patient samples tested showed undetermined or >35 amplification cycle numbers (Appendix A). The samples that showed a value for amplification did not show a single peak that matched with the positive control on the melt curve (Appendix A). These results indicated that the mRNA expression of HCMV IE1 was not detected in any of the patient samples analyzed, even if HCMV gB DNA was present. This indicates that the HCMV infection was latent in these tumors.

## 4. Discussion

Increasing evidence shows that HCMV infection is associated with breast cancer and metastasis [20,21]; however, the reported rate of positive detection of HCMV DNA/mRNA/proteins in breast tumors is highly variable in studies performed around the world, which ranges from 0 to 100% [19,41,42,43,44,45,46,47,48,49,50,51,52,53,54,55,56,57,58,61,62]. While this discrepancy could be partly caused by differences in active infection levels in different countries, it is also likely that the use of different detection methods could be a contributing factor. We compared real-time and gel-based PCR techniques for the detection of HCMV DNA and determined that a nested PCR, targeting the HCMV *gB* gene, was both sensitive and specific for analyzing human breast tumors. In contrast, using this same technique to detect the HCMV *IE1* gene led to non-specific results.

We therefore used nested PCR to measure HCMV gB DNA in breast tumors from a local Canadian patient population in addition to assessing the available matched serum samples for HCMV seropositivity. Patients with metastatic tumors were 5.61 or 5.27 times more likely to be HCMV seropositive or to have HCMV gB DNA-positive tumors, respectively, while the latter also had decreased relapse-free survival.

Detection of HCMV DNA in tissues normally relies on PCR; however, the reported results are highly variable. One explanation could be the variability and specificity of the methods used. Real-time PCR using the LightCycler, quantifies the target gene, while standard and nested PCR only detect the presence of the end products on a gel. We compared LC-PCR and nested PCR directly for the detection of HCMV gB DNA in our patient samples. Strict precautionary measures were taken when both PCR techniques were performed to avoid possible cross-contamination. Although the LC-PCR was very specific with the use of hybridization probes and allowed for real-time quantitative analysis of the viral load, nested PCR showed a 1000-fold higher sensitivity; however, the results from the nested PCR only indicated the presence of the viral DNA, not the amount, which limits information about the infection status. Several studies showing very little or no detection of HCMV DNA in breast tumors used detection methods similar to LC-PCR, which involved real-time analysis that was not the most sensitive detection method in our hands [45,47,51,52,53,54,55]; meanwhile, two other studies that used real-time PCR detected HCMV IE DNA in all 12 and 146 patient samples tested ([41,43], respectively). These same samples were all positive by IHC for HCMV IE or L proteins, which indicates an active infection with a high level of virus that likely exceeded the detection limit of real-time PCR [41,43]. In fact, one of these studies also showed the detection of HCMV mRNA in the tested tumors [43].

DNA detection must be both sensitive and specific to avoid false-positive results. In comparison to standard PCR, nested PCR uses two different sets of primers with one nested in the other and also uses two rounds of PCR reactions to enhance the amplification while ensuring its specificity. To confirm specificity, we performed Sanger DNA sequencing on the nested PCR products and demonstrated that only the expected product was obtained for HCMV gB. By contrast, nested PCR products obtained from targeting the HCMV *IE1* gene were not specific. This was surprising because the same HCMV IE1 primers and PCR conditions were described previously to be specific when the PCR products were sequenced [58]. El-Shinawi et al. detected HCMV IE1 DNA in 53.1% (26 out of 49) of tumors from non-inflammatory breast cancer patients with no detection in normal breast tissues from women free of cancer (the sample number was not specified) [58]. In a later study with more patients, the same authors reported HCMV IE1 DNA in 74% (67 out of 91) of non-inflammatory breast cancer patients and 89% (39 out of 44) of inflammatory breast cancer patients [50]. Although we followed the exact PCR detection method described in these studies, our PCR products were non-specific and resulted from the amplification of human genome sequences. We also tried to optimize the PCR reaction by testing a gradient of annealing temperatures based on what was reported, since the efficiency of PCR machines may be different; however, this did not decrease the non-specific products. A possible explanation for this discrepancy is the mismatch between the primers used in each study and the local circulating strains of HCMV. Genome variability of HCMV has been reported in congenitally infected infants [68], with the *IE1* gene potentially evolving, since it is a common target for CD8^+^ T cell responses [69]. Another possible explanation is that the HCMV DNA amount in the samples differed between their studies and ours, where non-specific reactions could overpower the target gene amplification if the level of viral genome present was very low. This is likely since the gels illustrated by El-Shinawi et al. also showed some non-specific bands, although these were very faint compared to the target gene product [58]. This suggestion is also supported in our study, since the HCMV positive control generated after in vitro infection of the human breast tissue, contained a high level of viral DNA that was able to produce a single strong band on the gel at the expected position. Nevertheless, the use of nested PCR to detect HCMV IE1 DNA was not a reliable method for analyzing patient samples in our hands. Instead, we used nested PCR targeting HCMV gB DNA, which was both sensitive and specific.

Previous studies showed a significantly higher percentage of breast tumors positive for HCMV DNA and/or proteins compared to normal breast tissues [19,43,44,45,57]. This could provide evidence that HCMV fulfills one of the criteria for being an oncovirus [70]; however, two other groups reported no significant differences in the detection of HCMV DNA in breast tumors versus normal breast tissues [48,52], which are compatible with our results. In our study, a high prevalence of HCMV DNA in breast tumors compared to normal breast tissues was not observed, although the low sample size for normal breast tissues could be statistically limiting. Compared to the seropositivity rate of 63% determined for this breast cancer patient population, HCMV gB DNA was only detected in 18% of the breast tumors. This result is compatible with other published studies [45,57]. Additionally, in our studies with mice that were latently infected with mCMV, we could only detect mCMV DNA in one or two but not all tissues examined including salivary gland, spleen, kidney, lung and breast tumors [63,71]. It is possible in this work that viral DNA was present in more tissues, but that it was below the level of detection.

We found no evidence for active or reactivated HCMV infections in the HCMV DNA-positive breast tumors or in breast tissue from women without cancer as shown by the lack of mRNA expression for HCMV IE, indicating that infection in these tissues was latent. This is surprising since the reactivation of CMV infection can be induced by inflammation, and we showed previously that there is a proinflammatory condition at the breast tumor site and in the adjacent adipose tissues [72]. Additionally, reactivation of CMV often occurs in differentiated macrophages and dendritic cells, which are present in breast tumors [73,74,75]. The seropositivity rate of 63% HCMV in our Canadian breast cancer patients is within the expected 40–70% infection rate for the adult population in North America [26]; however, this seropositivity is lower than in other countries where almost all of the patients are infected as determined through HCMV IgG testing [45] or the presence of HCMV DNA in the breast tumors [19,41,42,43,44]. In fact, HCMV mRNA was detectable in breast tumors in studies where 100% of the tumors examined were positive for HCMV DNA and/or proteins [42,43]. It is possible that there is a higher reactivation rate leading to the detection of mRNA in countries with very high HCMV infection rates. It is also possible that there was HCMV mRNA in the breast tumors from our patient population, but that the expression was very low and below the level of detection.

An important finding of this present study is that women who were seropositive for HCMV IgG regardless of the presence of gB DNA in their breast tumors were more likely to develop Stage IV metastatic breast cancer (OR 5.61, CI 1.09–28.75, *p* = 0.039). This was also true for breast cancer patients with HCMV gB DNA-positive tumors compared to all other patients with gB DNA-negative tumors regardless of HCMV status even without evidence of an active HCMV infection. More significantly, women who were HCMV seropositive and whose breast tumors were gB DNA-positive were more likely to develop Stage IV metastatic breast cancer with vascular involvement compared to those women who were HCMV seropositive but gB DNA-negative. These results are compatible with work where HCMV DNA and/or proteins are abundantly expressed in metastatic sentinel lymph nodes [41,43] and brain tissues [62] from breast cancer patients, indicating an association between infection and metastasis. In addition, our preclinical study shows that mice with latent mCMV infection developed more and larger metastatic lung nodules [63].

HCMV seropositivity increased with age in our study as expected [27] and was also associated with higher body mass index, which is also a risk factor for breast cancer [2]. A significantly higher incidence of infection was found in post- and peri-menopausal women compared to pre-menopausal women. This should be considered for follow-up and treatment options that are dependent on menopause status. Our study did not measure differences in the levels of HCMV IgG. One study in which almost all patients were HCMV seropositive, shows higher HCMV IgG levels in breast cancer patients compared to the control group [57], while another study shows no differences [76]. As well, HCMV-seropositive women with breast cancer who are less than 40 years of age have higher mean IgG levels than those without breast cancer. Higher IgG levels indicate a primary or reactivated infection that could therefore be considered a risk factor for breast cancer [77].

The presence of HCMV gB DNA in breast tumors in the present study was associated with worse outcomes in breast cancer patients. A study with low detection of HCMV DNA in breast tumors at less than 10% shows no viral association with clinical factors [56]. Another study with ~76% HCMV gB DNA detection in breast tumors shows that HCMV infection is associated with poor overall and relapse-free survival [61]. In our study of Canadian women, having HCMV gB DNA-positive breast tumors was associated with a reduced relapse-free survival, overall tumor recurrence and metastasis, but was not associated with reduced overall survival.

The presence of HCMV antigens in breast tumors is correlated with the lack of expression of ER, PR or HER2 in some studies [42,46], while others show a relationship with HER2 overexpression in HCMV-positive breast tumors [44]. In our study, HCMV seropositivity or the presence of gB DNA in the breast tumors was not associated with tumor subtypes; however, most tumors in this cohort were hormone-receptor positive. In experiments with cultured cells, we showed that the level of HCMV infectivity of different breast cancer cell lines was much lower than in fibroblasts. This did not depend on whether the breast cancer cells were triple negative or hormone receptor positive [59]. Instead, infection levels depended on the expression of PDGFRα. This receptor facilitates HCMV uptake in epithelial cells [78]. Despite this, there was no significant association between HCMV seropositivity or the presence of gB DNA in the breast tumors and the mRNA expression of PDGFRα in our patient samples; however, the proportion and characteristics of different cell types that constitute the breast tumors of patients could have a major impact on whether the tumors are positive or not for HCMV.

## 5. Conclusions

This study specifically evaluated two commonly used DNA detection methods for analyzing the presence of the HCMV genome in human breast tumors. Nested PCR was much more sensitive than real-time PCR and it detected DNA for HCMV gB with high specificity. By contrast, detection of HCMV IE1 DNA was not specific because of interference from the human genome. Using nested PCR, 18% of tumors from our Canadian breast cancer patients were positive for HCMV gB DNA whereas 63% of the breast cancer patient patients were seropositive for HCMV. HCMV seropositive patients were more likely to be older and to include post- and peri-menopausal women or those with a high body mass index. We directly related HCMV seropositivity to Stage IV metastatic breast cancer for the first time. HCMV infection in breast tumors was presumed to be latent because although viral DNA was detected, viral mRNA was not. These results are compatible with our mouse studies where latent mCMV infection increased the size and number of lung metastases [63]. Our present study shows that determining the seropositivity for HCMV and subsequent detection of HCMV gB DNA in breast tumors could identify breast cancer patients who are more likely to develop metastatic cancer and warrant special treatment to increase their relapse-free survival.

## Figures and Tables

**Figure 1 cancers-14-01148-f001:**
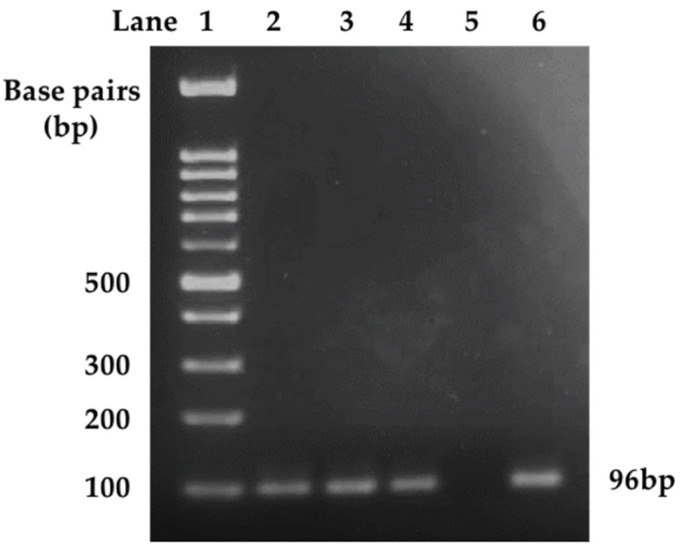
The detection limit of LightCycler PCR (LC-PCR) and nested PCR for HCMV *glycoprotein B (gB)* DNA. HCMV positive control was prepared in serial dilutions ranging from 1 to 1:10^5^. The samples were examined for the presence of HCMV *gB* DNA using LC-PCR (Table 1) and nested PCR in this Figure. The calculated concentration in Table 1 was determined by the LC-PCR program based on the previously established HCMV standards. Samples illustrated in lanes 1–6 are as follows: 100 base pair (bp) DNA ladder, HCMV 1:10^3^, 1:10^4^, 1:10^5^, negative control, positive control (HCMV 1:10). Undetermined cycle numbers are at least >40 and concentration cannot be calculated. 3.2. Specific Detection of HCMV gB but not HCMV *IE1* Using Analysis by Nested PCR of Human Specimens.

**Figure 2 cancers-14-01148-f002:**
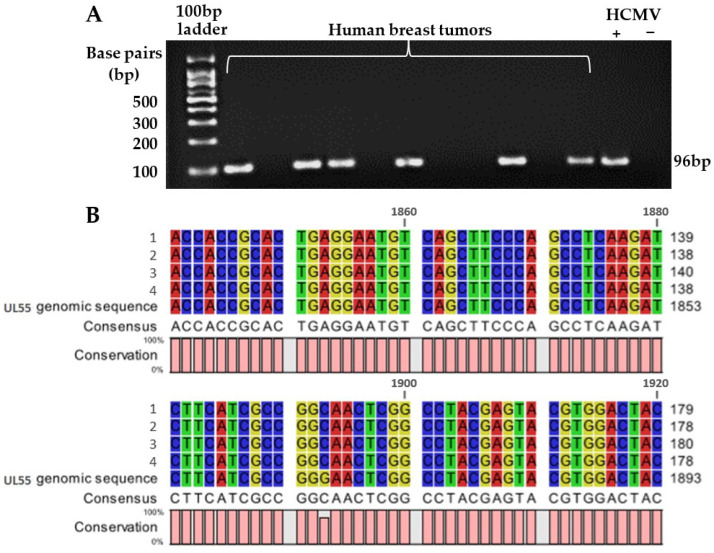
Specificity of HCMV glycoprotein (gB) DNA detection using nested PCR. PCR products for amplified HCMV *gB* gene from human breast tumor samples were visualized on 3% agarose gel stained with ethidium bromide, resulting in a band at 96 bp in the positive control and some human breast tumor samples (**A**). The PCR product at the 96 base-pair (bp) position was re-extracted from the gel and cloned for Sanger DNA sequencing, with the results aligned by comparing to the UL55 genomic sequence that encodes HCMV gB (**B**). Ten samples were analyzed and all aligned to UL55. Four of the samples were randomly chosen and illustrated in the sequence alignment map (**B**).

**Figure 3 cancers-14-01148-f003:**
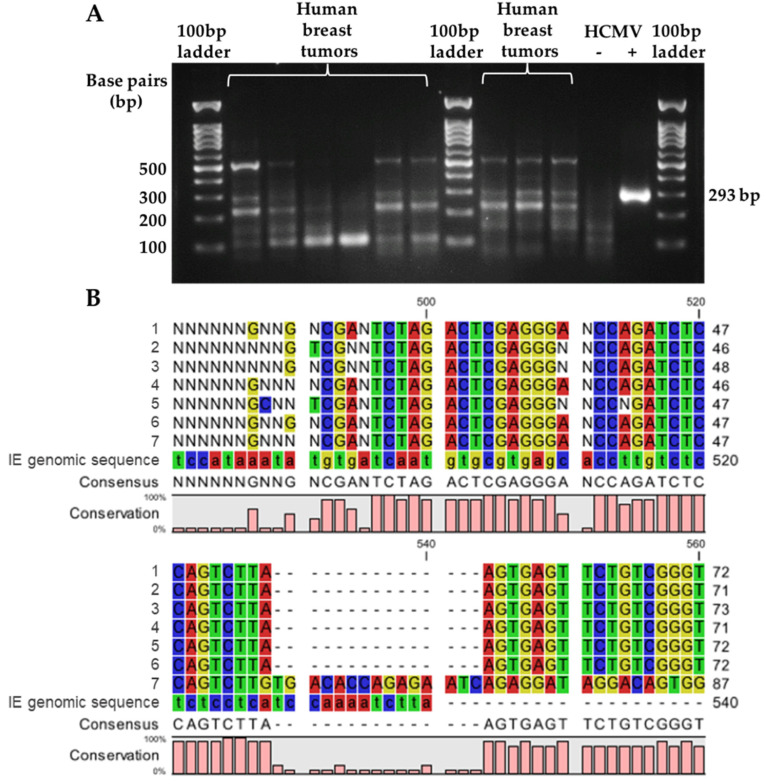
Specificity of HCMV immediate early 1 (IE1) DNA detection using nested PCR. PCR products for amplified HCMV *IE1* gene from human breast tumor samples were visualized on 1.5% agarose gel stained with ethidium bromide, resulting with a band at 293 base pair (bp) in the positive control (**A**). The PCR product at the 293 bp position was re-extracted from the gel for Sanger DNA sequencing, with the results aligned to the HCMV IE1 genomic sequence (**B**). Seven samples were analyzed and illustrated in the sequence alignment map (**B**).

**Figure 4 cancers-14-01148-f004:**
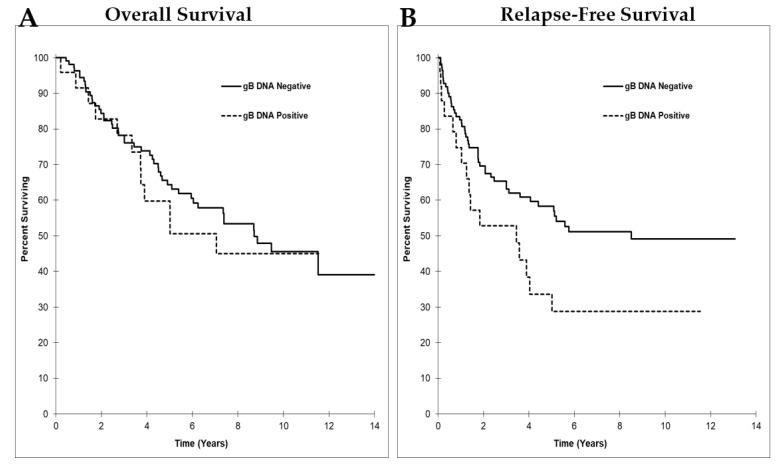
Kaplan–Meier survival curves based on HCMV gB DNA detection in breast tumors. (**A**) Overall survival and (**B**) Relapse-free survival. N = 136. *p*-values were calculated by the Chi-square analysis. *p*-value was considered significant if < 0.05. (**A**) *p* = 0.614 and (**B**) *p* = 0.039.

**Table 1 cancers-14-01148-t001:** HCMV positive control was prepared in serial dilutions ranging from 1 to 1:10^5^. The samples were examined for the presence of HCMV *gB* DNA using LC-PCR.

Sample	PCR Cycle Number	Expected Concentration (Copies/PCR Reaction)	Calculated Concentration (Copies/PCR Reaction)
Standard 1800 copies	29.72	1800	1480
Standard 180,000 copies	22.95	180,000	304,000
HCMV original	32.26	Unknown	200
HCMV 1:10	33.10	Unknown	104
HCMV 1:10^2^	38.18	Unknown	2
HCMV 1:10^3^	Undetermined	Unknown	NA
HCMV 1:10^4^	Undetermined	Unknown	NA
HCMV 1:10^5^	Undetermined	Unknown	NA
Negative control	Undetermined	Unknown	NA

NA = not available.

**Table 2 cancers-14-01148-t002:** Detection of HCMV DNA in tissues and IgG in serum of patients.

	Breast Tissues	All Breast Tumors	Serum	Breast Tumors from HCMV Seropositive Women
HCMV gB, *n* (%)				
Negative	9 (90)	111 (81.6)	NA	34 (57.6)
Positive	1 (10)	25 (18.4)		25 (42.4) *
HCMV IgG, n (%)				
Negative	NA	NA	35 (37.2)	NA
Positive			59 (62.8) *	

* There were ten patients with HCMV gB DNA-positive tumors who had no matching serum. These patients were assumed to be seropositive for HCMV. *p* = 0.691 comparing the number of HCMV gB-positive breast tissues to breast tumors was calculated by the Fisher’s exact test. * *p* value < 0.1 considered significant. NA = not applicable, gB = glycoprotein B.

**Table 3 cancers-14-01148-t003:** HCMV gB DNA negative and positive breast tumors: univariate analysis of patient characteristics.

Characteristics	HCMV gB− *n* = 111	HCMV gB+ *n* = 25	*p*-Value
Age, year (mean ± SD)	56 ± 13	57 ± 13	0.596
Menopause status, n (%)			0.482
Pre, *n* = 38	32 (29.9)	6 (24.0)	
Post, *n* = 85	69 (64.5)	16 (64.0)	
Peri, *n* = 9	6 (5.6)	3 (12.0)	
Body mass index (mean ± SD)	29.0 ± 6.8	31.0 ± 8.8	0.223
Deceased, *n* (%)			0.665
No, *n* = 76	63 (56.8)	13 (52.0)	
Yes, *n* = 60	48 (43.2)	12 (48.0)	
Recurrence, *n* (%)			0.060
No, *n* = 72	63 (56.8)	9 (36.0)	
Yes, *n* = 64	48 (43.2)	16 (64.0)	
Time to recurrence event, days from diagnosis (median and IQR)	1132 (41–4774)	859 (54–4216)	0.424
Time to recurrence event, days from surgery (median and IQR)	1132 (40–4774)	673 (34–4216)	0.237
Tumor grade, *n* (%)			0.616
Low, *n* = 45	36 (33.6)	9 (39.1)	
High, *n* = 85	71 (66.4)	14 (60.9)	
Positive lymph nodes, *n* (mean ± SD)	4 ± 5	4 ± 6	0.619
Size of largest lymph nodes, cm(mean ± SD)	1.58 ± 0.83	1.57 ± 0.93	0.959
Tumor stage, *n* (%)			0.007
I–III, *n* = 116	99 (89.2)	17 (68.0)	
IV = metastasis, *n* = 20	12 (10.8)	8 (32.0)	
ER, *n* (%)			0.745
Negative, *n* = 18	14 (12.8)	4 (16.0)	
Positive, *n* = 116	95 (87.2)	21 (84.0)	
PR, *n* (%)			0.745
Negative, *n* = 47	37 (33.9)	10 (40.0)	
Positive, *n* = 87	72 (66.1)	15 (60.0)	
HER2, *n* (%)			0.745
Negative, *n* = 103	85 (76.6)	18 (75.0)	
Positive, *n* = 32	26 (23.4)	6 (25.0)	
Tumor subtypes, *n* (%)			0.998
Triple negative, *n* = 5	4 (3.6)	1 (4.2)	
Luminal A, *n* = 97	80 (72.7)	17 (70.8)	
Luminal B, *n* = 21	17 (15.5)	4 (16.7)	
HER2 Enriched, *n* = 11	9 (8.2)	2 (8.3)	
Vascular invasion, *n* (%)			0.225
No, *n* = 53	46 (42.6)	7 (29.2)	
Yes, *n* = 79	62 (57.4)	17 (70.8)	
Tumor PDGFRα mRNA (mean ± SD)	7.92 ± 6.90	6.53 ± 5.15	0.374

*p*-values were calculated by Chi-square analysis or the Fisher’s exact test if *n* < 5 in a category. *p*-value was considered significant if <0.1 and these values are bolded. ER: estrogen receptor; gB: glycoprotein B; HER2: human epidermal growth factor receptor 2; PDGFRα: platelet-derived growth factor receptor alpha; PR: progesterone receptor.

**Table 4 cancers-14-01148-t004:** HCMV IgG seronegative and seropositive patients: univariate analysis of patient characteristics.

Characteristics	IgG− (*n* = 35)	IgG+ * (*n* = 59)	*p*-Value
Age, year (mean ± SD)	51 ± 13	57 ± 12	**0.017**
Menopause status, *n* (%)			**0.012**
Pre, *n* = 29	17 (50.0)	12 (21.1)	
Post, *n* = 55	16 (47.1)	39 (68.4)	
Peri, *n* = 7	1 (2.9)	6 (10.5)	
Body mass index (mean ± SD)	27.5 ± 6.0	30.0 ± 7.4	**0.093**
Deceased, *n* (%)			0.617
No, *n* = 56	22 (62.9)	34 (57.6)	
Yes, *n* = 38	13 (37.1)	25 (42.4)	
Recurrence, *n* (%)			0.148
No, *n* = 50	22 (62.9)	28 (47.5)	
Yes, *n* = 44	13 (37.1)	31 (52.5)	
Time to recurrence event, days from diagnosis (median and IQR)	1509 (126–4774)	1128 (54–4216)	0.375
Time to recurrence event, days from surgery (median and IQR)	1509 (126–4774)	1007 (34–4216)	0.290
Tumor grade, *n* (%)			0.343
Low, *n* = 32	10 (29.4)	22 (39.3)	
High, *n* = 58	24 (70.6)	34 (60.7)	
Positive lymph nodes, *n* (mean ± SD)	3 ± 4	4 ± 5	0.255
Size of largest lymph nodes, cm (mean ± SD)	1.52 ± 0.74	1.50 ± 0.79	0.941
Tumor stage, *n* (%)			0.157
I–III, *n* = 79	32 (91.4)	47 (79.7)	
IV = metastasis, *n* = 15	3 (8.6)	12 (20.3)	
ER, *n* (%)			0.741
Negative, *n* = 11	3 (8.8)	8 (13.8)	
Positive, *n* = 81	31 (91.2)	50 (86.2)	
PR, *n* (%)			0.937
Negative, *n* = 32	12 (35.3)	20 (34.5)	
Positive, *n* = 60	22 (64.7)	38 (65.5)	
HER2, *n* (%)			0.644
Negative, *n* = 72	28 (80.0)	44 (75.9)	
Positive, *n* = 21	7 (20.0)	14 (24.1)	
Tumor subtypes, *n* (%)			0.220
Triple negative, *n* = 4	2 (5.9)	2 (3.4)	
Luminal A, *n* = 67	25 (73.5)	42 (72.4)	
Luminal B, *n* = 15	7 (20.6)	8 (13.8)	
HER2 Enriched, *n* = 6	0 (0)	6 (10.3)	
Vascular invasion, *n* (%)			0.981
No, *n* = 40	15 (44.1)	25 (43.9)	
Yes, *n* = 51	19 (55.9)	32 (56.1)	
Tumor PDGFRα mRNA (mean ± SD)	7.8 ± 6.62	7.08 ± 5.90	0.591

* Ten patients with HCMV gB DNA positive tumors but no matching serum available were assumed to be seropositive for HCMV and included in the IgG+ group. *p*-values were calculated by Chi-square analysis or the Fisher’s exact test if n <5 in a category. *p*-value was considered significant if < 0.1 and these values are bolded. ER: estrogen receptor; gB: glycoprotein B; HER2: human epidermal growth factor receptor 2; PDGFRα: platelet-derived growth factor receptor alpha; PR: progesterone receptor.

**Table 5 cancers-14-01148-t005:** HCMV IgG seronegative and seropositive patients with and without detection of HCMV gB in breast tumors: univariate analysis for patient characteristics.

Characteristics	IgG+/gB− *n* = 34	IgG+/gB+ * *n* = 25	*p*-Value
Age, year (mean ± SD)	57 ± 12	57 ± 13	0.985
Menopause status, *n* (%)			0.818
Pre, *n* = 12	6 (18.8)	6 (24.0)	
Post, *n* = 39	23 (71.9)	16 (64.0)	
Peri, *n* = 6	3 (9.4)	3 (12.0)	
Body mass index (mean ± SD)	29.3 ± 6.2	31.0 ± 8.8	0.406
Deceased, *n* (%)			0.453
No, *n* = 34	21 (61.8)	13 (52.0)	
Yes, *n* = 25	13 (38.2)	12 (48.0)	
Recurrence, *n* (%)			0.131
No, *n* = 28	19 (55.9)	9 (36.0)	
Yes, *n* = 31	15 (44.1)	16 (64.0)	
Time to recurrence event, days from diagnosis (median and IQR)	1130 (175–4205)	859 (54–4216)	0.453
Time to recurrence event, days from surgery (median and IQR)	1115 (87–4205)	673 (34–4216)	0.276
Tumor grade, *n* (%)			0.984
Low, *n* = 22	13 (39.4)	9 (39.1)	
High, *n* = 34	20 (60.6)	14 (60.9)	
Positive lymph nodes, *n* (mean ± SD)	4 ± 5	4 ± 6	0.671
Size of largest lymph nodes, cm (mean ± SD)	1.45 ± 0.68	1.57 ± 0.93	0.612
Tumor stage, n (%)			**0.0995**
I–III, *n* = 47	30 (88.2)	17 (68.0)	
IV = metastasis, *n* = 12	4 (11.8)	8 (32.0)	
ER, *n* (%)			0.715
Negative, *n* = 8	4 (12.1)	4 (16.0)	
Positive, *n* = 50	29 (87.9)	21 (84.0)	
PR, *n* (%)			0.442
Negative, *n* = 20	10 (30.3)	10 (40.0)	
Positive, *n* = 38	23 (69.7)	15 (60.0)	
HER2, *n* (%)			0.897
Negative, *n* = 72	26 (76.5)	18 (75.0)	
Positive, *n* = 21	8 (23.5)	6 (25.0)	
Tumor subtypes, *n* (%)			0.923
Triple negative, *n* = 2	1 (2.9)	1 (3.4)	
Luminal A, *n* = 42	25 (73.5)	17 (72.4)	
Luminal B, *n* = 8	4 (11.8)	4 (13.8)	
HER2 Enriched, *n* = 6	4 (11.8)	2 (10.3)	
Vascular invasion, *n* (%)			**0.057**
No, *n* = 25	18 (54.5)	7 (29.2)	
Yes, *n* = 32	15 (45.5)	17 (70.8)	
Tumor PDGFRα mRNA (mean ± SD)	7.45 ± 6.42	6.53 ± 5.15	0.579

* Ten patients with HCMV gB DNA positive tumors but no matching serum available were assumed to be seropositive for HCMV and included in this analysis. *p*-values were calculated by Chi-square analysis or the Fisher’s exact test if n < 5 in a category. *p*-value was considered significant if < 0.1 and these values are bolded. ER: estrogen receptor; gB: glycoprotein B; HER2: human epidermal growth factor receptor 2; PDGFRα: platelet-derived growth factor receptor alpha; PR: progesterone receptor.

**Table 6 cancers-14-01148-t006:** Multivariate analysis of patient characteristics identified by univariate analysis to determine the odds ratio of being HCMV gB DNA and/or IgG positive.

	Variable	OR	95% CI	*p*-Value
gB status (*n* = 136)	Stage I–III Stage IV	5.27	1.77–15.67	**0.003**
	Pre-menopause			
	Post-menopause	1.31	0.45–3.84	0.621
	Peri-menopause	3.95	0.73–21.44	0.112
IgG * status (*n* = 94)	Stage I–III Stage IV	5.61	1.09–28.75	**0.039**
	Pre-menopause			
	Post-menopause	3.83	1.43–10.29	**0.008**
	Peri-menopause	11.42	1.18–110.31	**0.035**
IgG/gB* status (*n* = 57)	Stage I–III Stage IV	3.48	0.88–13.78	0.076
	Pre-menopause			
	Post-menopause	0.80	0.21–3.11	0.748
	Peri-menopause	1.50	0.20–11.42	0.697

* Ten patients with HCMV gB DNA-positive tumors but no matching serum available were assumed to be seropositive for HCMV and included in the IgG+ group. *p*-values were calculated by multiple logistic regression analysis. *p*-value was considered significant if < 0.05 and these values are bolded. OR: odds ratio; CI: confidence interval.

## Data Availability

All relevant data for the patients in this study is provided in the Tables, Figures and Appendix A. All patient samples were deidentified to preserve patient confidentiality.

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
