# Peer review of "Human Cytomegalovirus Seropositivity and Viral DNA in Breast Tumors Are Associated with Poor Patient Prognosis"

_cancers, 2022, doi:10.3390/cancers14051148_

Round 1

Reviewer 1 Report

The present manuscript indicates that HCMV seropositivity and HCMV-DNA-positivity in breast tumours may potentially be used as prognostic factors in the clinic. Thus, latent HCMV-infections may eventually become a target in breast cancer patients in the future (at least for subgroups of patients, like triple-negative BC etc.).

I have only one comment regarding the presentation of the results. I understand that the authors present here follow-up data for patients treated for early BC. If this is correct, I recommend to use the term "relapse free survival" (time from surgery to relapse) and not progression-free survival (that is used for the metastatic setting only).

All in all, very interesting research and results that should be of interest for everyone treating breast cancer.

Author Response

We thank the reviewer for the complimentary comments concerning our paper and the suggestion,  which we really appreciate. We agree to replace "progression-free survival" by "relapse-free survival" throughout the paper because this is more accurate and have done so. 

Reviewer 2 Report

There is controversy on the role or presence of Cytomegalovirus (CMV) in some cancers, including breast cancer.  The authors have previously demonstrated that CMV infection in the setting of a mouse model of metastatic breast cancer, may influence the metastatic process.  Here, the authors present a very rigorous analysis of CMV detection with nested PCR to identify truly CMV-infected breast cancer tumors from human biopsy specimens. While CMV expression was not detected in tumors, the presence of CMV DNA and seropositivity to CMV were statistically associated with increased breast cancer metastasis - corroborating the findings in their mouse model. 

I believe the authors should be commended for their careful analysis of CMV by PCR.  Overall their methodologies appear solid and reproducible.  

The fact that CMV "latent" infection and evidence of CMV exposure  positively influence the metastatic potential for human breast cancer could be an important discovery. Perhaps the "latent" CMV actually is expressing some genes at low levels. As the authors note, other investigators' work has suggested CMV gene products could influence tumor growth and immune evasion. Most striking is the PFS data for patients whose tumors were CMV gB DNA positive vs negative.  

Further work will need to be done to determine the mechanism through which CMV may potentiate metastasis.  Work by Shenk's group showed that CMV promotes a mesenchyal to epithelial transition in breast cancer cells, which is required for metastatic growth. 

I believe these data will increase further research into this novel and potentially important study of  how the tumor microbiome can influence metastatic potential. 

Author Response

We thank this reviewer for the positive comments about our work, which we really appreciate. We agree totally that further work will need to be done to determine the mechanism through which CMV may potentiate metastasis.  We are indeed following up studies by the Shenk group, which showed that CMV promotes a mesenchymal to epithelial transition in breast cancer cells, which is required for metastatic growth. Hopefully, these studies will provide further mechanistic explanation for the cause of CMV-induced metastasis.